# Intensive Treatment of Chronic Pain and PTSD: The PATRIOT Program

**DOI:** 10.3390/bs14111103

**Published:** 2024-11-16

**Authors:** John D. Otis, Jonathan S. Comer, Terence M. Keane, Erica Checko (Scioli), Donna B. Pincus

**Affiliations:** 1Department of Veterans Affairs, VA Boston Healthcare System, Boston, MA 02130, USA; terry.keane@va.gov (T.M.K.);; 2Department of Psychological and Brain Sciences, Center for Anxiety and Related Disorders at Boston University, Boston, MA 02215, USA; dpincus@bu.edu; 3Mental Health Interventions and Novel Therapeutics (MINT) Program, Florida International University, Miami, FL 33199, USA; jocomer@fiu.edu; 4Department of Psychiatry, Boston University, Boston, MA 02118, USA

**Keywords:** post-traumatic stress disorder, chronic pain, cognitive behavioral therapy

## Abstract

Military combat can result in the need for comprehensive care related to both physical and psychological trauma, most commonly chronic pain and post-traumatic stress disorder (PTSD). These conditions tend to co-occur and result in high levels of distress and interference in everyday life. Thus, it is imperative to develop effective, time-efficient treatments for these conditions before they become chronic and resistant to change. We developed and pilot-tested the Pain and Trauma Intensive Outpatient Treatment (PATRIOT) Program, a brief, intensive (3 weeks, six sessions) integrated chronic pain and PTSD treatment. An overview and session-by-session outline of the PATRIOT Program is provided, followed by results from the first pilot evaluation of the PATRIOT Program’s feasibility, acceptability, and preliminary efficacy in a sample of eight participating Veterans with comorbid chronic pain and PTSD. There were no treatment dropouts. At post-treatment, there were significant reductions in PTSD symptoms based on the Clinician-Administered Assessment of PTSD (CAPS). Pain and catastrophic thinking also decreased from pre- to post-treatment. With continued investigations and support, the PATRIOT Program may offer a brief, cost-effective, and more easily accessible treatment option for individuals who could benefit from learning skills to manage pain and PTSD more effectively.

## 1. Introduction

Military combat can result in the need for comprehensive care related to both physical and psychological trauma, most commonly chronic pain and post-traumatic stress disorder (PTSD). The rates of chronic pain and PTSD reported by Veterans receiving treatment in the VA healthcare system are very high. A recent study found that, of the 5,846,453 service users of the VA in 2023, a total of 2,091,391 (35.8%) met the criteria for chronic pain and 850,191 (14.5%) met the criteria for PTSD. Furthermore, 21.6% of those with chronic pain also had PTSD, and over half (53.2%) of those with PTSD also met the criteria for chronic pain (*n*  =  452,113) [1]. Even higher rates of pain and PTSD have been observed in specialty care clinics. For example, a study of 340 OEF/OIF Veterans found that chronic pain and PTSD were present in 81.5% and 68.2% of Veterans, respectively [2]. In a study of 50 Operation Enduring Freedom/Operation Iraqi Freedom (OEF/OIF) injured service members treated at a level 1 polytrauma rehabilitation center responsible for addressing the healthcare needs of more seriously injured Veterans, 96% reported a pain problem and 78% reported experiencing mental health problems, with the most common problem being PTSD (44%) [3]. Taken together, these studies demonstrate the high co-prevalence rates of these conditions.

An increasing body of research has focused on developing a greater understanding of the interactions between chronic pain and PTSD [4,5]. Research indicates that patients with chronic pain and PTSD experience more intense pain and affective distress [6], higher levels of life interference [7], and greater disability [8] than patients with either pain or PTSD alone. Veterans who have both chronic pain and PTSD endorse significantly higher levels of maladaptive coping strategies and beliefs about pain (i.e., greater catastrophizing and emotional impact on pain; less control over pain) when compared to Veterans with chronic pain alone [9]. Not surprisingly, Veterans with chronic pain and PTSD also demonstrate worse pain and psychological outcomes following pain treatment when compared to those with pain alone [10].

Given the current theoretical models describing the potential mechanisms through which chronic pain and PTSD might be maintained [11,12], and the similarities among the psychological treatment approaches for both conditions, Otis and colleagues pursued a line of research aimed at developing a more effective and integrated approach to treating comorbid chronic pain and PTSD. With funding provided by the Department of Veterans Affairs, Otis and colleagues developed and tested a 12-session treatment protocol that was informed by current theory and combined essential elements of cognitive processing therapy (CPT) for PTSD [13] and cognitive behavioral therapy (CBT) for chronic pain [14]. The results of a preliminary study supported the initial feasibility of a 12-session integrated pain and PTSD treatment approach [15]; however, some limitations were noted. Specifically, the treatment was reported by some participants to be too lengthy. Veterans often have multiple health and financial issues or competing appointments, which can interfere with their ability to attend weekly therapy appointments. Some Veterans reported wanting more immediate relief from the significant impairment and disruption of everyday activities that had resulted from their pain and PTSD. Veterans reported not wanting to spend a protracted amount of time in the VA medical center. They reported wanting to return to work, to participate in more pleasant activities, and to have something “functional” to do in their lives, but their pain and PTSD were interfering. In fact, several Veterans inquired about whether a “quicker” form of therapy was available to facilitate their re-entry and involvement in daily activities. Further, it was observed by the study therapists that missed appointments by participants would often create gaps in treatment that reduced the retention of information from the previous session and decreased the participants’ ability to effectively deploy skills when needed outside the session. This feedback prompted interest in refining the content and format of the 12-session integrated pain and PTSD treatment so that it could be delivered in a more “intensive” or time-efficient manner. 

Numerous studies have shown support for brief, intensive treatments for psychological disorders, especially anxiety disorders such as panic disorder and specific phobias [16,17,18,19,20]. In a review of brief cognitive behavioral therapies, Hazlett-Stevens and Craske [18] detailed various benefits of intensive treatments, such as “increased cost-effectiveness” and “rapid treatment gains” with the potential to “facilitate increased motivation”. Taken together, research suggests that intensive therapies may be as effective as non-abbreviated therapies in the treatment of many debilitating conditions and may have a variety of advantages as compared to standard-length treatments.

To address the unique needs of Veterans suffering from comorbid chronic pain and PTSD in a more efficient and acceptable manner, the Pain and Trauma Intensive Outpatient Treatment (PATRIOT) Program was developed. The PATRIOT Program is an adaptation of the initial 12-session integrated pain and PTSD treatment approach [12], but instead entails only six sessions delivered across 3 weeks. The present paper details the development of the PATRIOT Program, provides an overview of the Program, and presents an open pilot evaluation that offers the first examination of the feasibility, acceptability, and initial efficacy of the PATRIOT Program. It was hypothesized that (1) the PATRIOT Program would be feasible to implement in the VA healthcare system; (2) Veterans receiving treatment would report high rates of satisfaction and demonstrate high rates of therapy attendance; and (3) participants would report reductions in pain, PTSD symptoms, interference, depressive symptoms, and pain-related catastrophizing. 

## 2. Materials and Methods

### 2.1. The PATRIOT Program

The Pain and Trauma Intensive Outpatient Treatment (PATRIOT) Program was designed to be delivered in 6 sessions, twice a week, for 3 weeks. Each session is 90 min in duration and includes elements of CBT for pain [14] and CPT for PTSD [13]. The truncated format of the treatment was modeled after existing intensive psychological treatments that had gained much empirical support [18,20].

While the treatment’s development was informed by our previous research, there were a number of elements that were unique. It was observed by the study therapists from our initial 12-session program [15] that the standard 12-session integrated pain–PTSD treatment included cognitive elements (i.e., trust/safety, intimacy, self-esteem) that did not resonate with all Veterans. For example, Veterans who did not have issues with intimacy seemed less engaged when discussing material covering this topic. Thus, it was decided that, while the PATRIOT Program would continue to have a strong cognitive component, the information presented would be tailored to the specific needs of each Veteran. For example, based on the information presented in an “Impact Statement” written by the participants and reviewed in Session 2 of the treatment, the PATRIOT therapist and patient now select a treatment “module” that most closely relates to the “stuck-point” or issue in which the participant is having the most cognitive challenges. Three PATRIOT modules were created, including Module 1 = Anger Management, Module 2 = Power and Control, and Module 3 = Safety and Trust. This modification allowed the treatment to more closely and flexibly approximate clinical practice. More recent research indicating a high prevalence of sleep problems in Veterans with pain, PTSD, and TBI prompted the inclusion of a stronger sleep hygiene component in the PATRIOT Program and skills to address nightmares. Finally, a greater emphasis was placed on teaching means of developing social support, making friends, and re-engaging with friends and family in a healthy way.

Goal setting and achievement are integrated into every PATRIOT session. Three types of goals are set in therapy: overall goals, behavioral goals, and homework goals. Overall goals, which are set in Session 1, are goals that can be achieved by the end of treatment and are measurable (e.g., walking daily on the treadmill, going to the gym, spending 30 min a day organizing the office) rather than vague (e.g., be a better person, decrease pain). Overall goals can be modified during therapy if (1) the patient achieves the goal and would like to set another or (2) the patient changes his/her mind about what he/she would like to work towards. Behavioral goals are small achievable goals set after each therapy session that help the patient to work towards an overall goal. For example, if the overall goal is to walk 30 min daily on the treadmill, a behavioral goal might be to walk 10 min daily on the treadmill. The behavioral goal can be increased in small steps until the overall goal is reached. Homework goals are associated with the material covered during each session. For example, in addition to having the behavioral goal of walking 10 min a day, a patient may have the homework goal of performing diaphragmatic breathing daily for 10 min or of restructuring three maladaptive thoughts per day. Following the presentation of the content of each session, behavioral and homework goals are developed. Each PATRIOT session begins with a review of the behavioral goals and homework assigned from the previous session using a goal completion form.

A PATRIOT therapist manual and a patient workbook were created in order to standardize treatment delivery. The therapist manual contains a bulleted list of essential treatment elements to be addressed in each session, all of the content of the treatment organized by session, and sample text that could be used by therapists to relay key concepts. Patients are given a workbook at the start of treatment that is organized by session and contains patient-friendly versions of all of the content and homework assignments. Table 1 provides an overview of the session-by-session structure of the PATRIOT Program.

### 2.2. The PATRIOT Program Outline by Session

#### 2.2.1. Session 1: Making the Connection

The goals of the first PATRIOT session are to present the treatment rationale (education on the factors that contribute to the development and maintenance of chronic pain and PTSD) and review elements connecting pain and PTSD, including negative thinking and avoidance. Reducing avoidance is a theme that is incorporated throughout treatment, rather than just in a particular therapy session. For example, a clinician can illustrate to the patient how avoiding reminders of a traumatic event, or avoiding potentially physically painful activities, prevents the patient from learning that these events do not need to be feared. Avoidance also prevents individuals from being fully involved in pleasurable activities of daily living, such as being around family and friends. Providing a rationale for addressing avoidance can help individuals to become more open and comfortable with setting therapy goals that include activity scheduling and exposure-based therapies. Topics that are discussed in relation to pain include the chronic pain cycle, which reviews how the experience of chronic pain can contribute to increased physical disability, and emotional distress, while topics related to PTSD include the fight or flight response. Next, patients are taught how to relax using diaphragm breathing. Breathing is taught because it is an effective way to relax, and practicing this helps people to become better observers of their own thoughts. In collaboration with the therapist, patients create overall behavioral goals for treatment, which include goals that they could reasonably expect to achieve over the next 5 sessions, such as starting an exercise program or re-engaging in a hobby or enjoyable activity. As part of their homework, participants are asked to write an “Impact Statement” of at least 1 page in length with instructions to “Consider what it means to you that you have chronic pain and/or that you experienced the traumatic event. What effects have these conditions had on your beliefs about yourself, others, and the world?” 

#### 2.2.2. Session 2: Cognitive Restructuring

The first goal of this session is to review the Impact Statement written by the patient. While listening to the patient read the Impact Statement, the therapist looks for thoughts that may interfere with the complete processing of the traumatic event, increase negative emotions, and/or result in increased avoidance related to the pain or trauma. If a patient does not complete the homework or forgets to bring it in, he/she is asked to complete the assignment during the session. After listening to the Impact Statement, the therapist reviews the major themes that were noticed and suggests that addressing these will be the focus of treatment. The therapist normalizes the impact of pain and PTSD on the patient’s life, but also begins to instill the idea that there may be other ways to think about what happened to the patient. The remainder of the session is spent discussing automatic thoughts, cognitive errors, and how to restructure thoughts. Homework for this session is to practice cognitive restructuring daily.

#### 2.2.3. Session 3: Focused Cognitive Restructuring

The session begins with a review of the cognitive restructuring homework from the previous session. After reviewing the patient’s homework and ensuring that they understand how to restructure their thoughts, the focus of this PATRIOT session returns to the Impact Statement to discuss the major themes of the thoughts discussed.


*Therapist: Let’s go back to the Impact Statement you read last session. From what you read, it sounds like thoughts related to (insert thought here) have really impacted your life (identify them). Does this sound correct? Ok, let’s talk more about these areas. (If not accurate, therapist clarifies with patient and identifies the relevant areas).*


Based on this conversation, the therapist and the patient work together to identify an area in which they would like to practice more focused cognitive restructuring. There are 3 treatment modules that can be selected, including Module 1 = Anger Management, Module 2 = Power and Control, and Module 3 = Safety and Trust. Module 1 teaches patients how anger can interact with the experiences of pain and PTSD. Patients are taught how to develop an awareness of their triggers for anger, to modify the internal process that can moderate anger (i.e., relaxation, perspective taking, restructuring), and ways to respond to others in an assertive yet non-aggressive manner. Module 2 is appropriate for the patient for whom the experience of pain and PTSD has resulted in distorted beliefs in their ability to control aspects of their life, as well as the extent to which power or control is in the hands of other people. For example, patients may believe that if they do not control everything in their life, then they are completely out of control. Patients are taught to identify these types of thoughts and develop more adaptive coping statements. Module 3 focuses on safety and trust and is appropriate for patients who have difficultly trusting their judgment or decision-making skills. These types of beliefs could develop if a patient believes that they made a decision that led to a traumatic event or an injury (e.g., “If I had looked closer I could have noticed the IED (improvised explosive device) on the side of the road and I wouldn’t have been hurt”). In the event that another person made a decision that led to the traumatic event or placed him/her in harm’s way, it can disrupt their trust of people in authority, including physicians, psychologists, and government officials.

#### 2.2.4. Session 4: Sleep and Relaxation Training

This session focuses on teaching patients ways to relax and improve sleep. Sleep hygiene techniques related to sleep timing, behaviors, bedroom environments, ingestion, and mental control are reviewed with the patient, and a checklist of adaptive sleep behaviors is assigned to follow for homework. Given that nightmares are a common feature of PTSD, an additional component of imagery rehearsal therapy (IRT) and exposure, relaxation, and rescripting therapy (ERRT) is included in this session [21]. The Veteran is asked to recall a commonly experienced nightmare and to consider the themes of power, control, and trust in the nightmare. For example, if a patient reported that they felt that they lacked power and control in the nightmare, they would be asked to rehearse new imagery during the day where they had more power and control. The patient is asked to write out the rescripted dream, read it aloud, and rehearse it daily as a homework assignment, along with a sleep hygiene routine and progressive muscle relaxation (PMR). For example, one participant reported experiencing a dream in which, after breaking through a door, he was immediately engaged in a firefight with the enemy and surrounded by muzzle flashes. Although the enemy was neutralized and the patient was reunited with his unit, he was stuck reliving the moments of combat in his dreams. As part of his therapy, he was asked to imagine the scene, move beyond the firefight, and focus on the positive feelings that he had when he joined his unit. By rehearsing this situation during the day and changing the storyline, he was able to change the basis for the nightmare.

#### 2.2.5. Session 5: Activity Pacing and Pleasant Activity

This session is focused on teaching participants to be more active and engage in enjoyable activities without overdoing it. This is an important skill because it is very common for people to think that they have to “work through” their pain in order to get things done. However, this approach can result in increased pain for days afterwards and the increased use of pain medication to compensate for pain. There is often some resistance when it is suggested to Veterans to take things at a slower pace—many Veterans are reluctant to admit when they need to slow down. Veterans report being trained to think that they should not stop working on a task until it is completed. The key to activity pacing is explaining that if a person paces appropriately, they will actually be able to get more things done in the long run. It is also common that when Veterans experience pain or PTSD, they isolate themselves from friends and stop doing the things that they once found the most enjoyable. Once removed from reinforcing activities and positive social situations, patients often become more depressed, which can further contribute to their disability. Pleasant activity scheduling involves identifying activities, setting reasonable goals and pacing, scheduling them into the week, and addressing procrastination and avoidance. 

#### 2.2.6. Session 6: Social Support and Integrating Skills into Everyday Life 

The final PATRIOT session focuses on the development of social support. The availability of social support and personal hardiness are protective factors with the potential to provide resiliency to those who have chronic pain or who have experienced a traumatic event. This session reviews the steps for developing a social support network, including how to choose friends, being proactive, being a good listener, and staying in touch. The session ends with an emphasis on the continued integration of new skills into everyday life and problem-solving what to do if the patient experiences a relapse of symptoms of chronic pain or PTSD. 

### 2.3. Open Pilot Series Method

The present open pilot series evaluation (N = 8) provides the first insights into the initial feasibility, acceptability, and preliminary efficacy of the PATRIOT Program.

#### 2.3.1. Participants

The study procedures were approved by the local VA Research and Development Committee and the Institutional Review Board prior to initiation, and all participants provided informed consent. Participants were recruited through the placement of advertisements in patient care areas in a Northeastern Department of Veterans Affairs (VA) medical center. Eligible participants were all U.S. Veterans (age 18 or over). Patients were eligible to participate if they had constant pain of at least three months’ duration with a neurologic or musculoskeletal etiology and met the DSM-IV diagnostic criteria for PTSD. Stability of pain medication and anxiety medication was required for 2 months prior to study entry. Participants were required to not participate in any new therapies for pain or PTSD in order to maintain the integrity of the treatment and to minimize the heterogeneity of the treatment received. Given that the treatment contained elements from both CBT for pain [14] and cognitive processing therapy (CPT) for PTSD [13], Veterans who had completed a trial of CBT for pain or CPT in the past year were not eligible to participate. Patients with a life-threatening/acute physical illness, a current substance use disorder, psychosis or suicidal intent, and individuals seeking surgical pain intervention were excluded. 

A total of 18 Veterans were consented to participate; however, 8 participants did not meet the criteria for PTSD. Of the remaining 10 participants, 2 were unable to be located after participating in the pre-treatment assessment and never began the treatment program. A total of 8 participants who met the diagnostic criteria for both chronic pain and PTSD were seen as part of this study (see Table 2 for demographic characteristics). Three participants served in supportive or combat roles in Vietnam, 2 participants served in non-combat roles, and 3 served in supportive or combat roles in OEF/OIF or the Gulf War. The participants consisted of 7 males and 1 female; the average age of the participants was 53.6 (SD = 18.3). Half of the participants were Caucasian non-Hispanic. All participants identified 2 or more pain sites, with the most common sites being shoulder, back, and knee pain. 

#### 2.3.2. Measures

Structured Clinical Interview for DSM-IV (SCID): SCID-I [22]: Alcohol and Drug, Mood Disorders, Anxiety Disorders, Somatoform, and Psychotic Screen. SCID-II: Anti-social Personality Disorder, Borderline Personality Disorder. The SCID is a clinician-administered interview used to assess for DSM-IV diagnoses. It has been successfully used across diverse clinical populations and is currently in widespread use in both clinical and research capacities, where it is generally considered the gold standard.

Clinician-Administered PTSD Scale (CAPS) [23]: This 30-item structured interview is based on the DSM-IV and is designed to assess both the 17 symptoms of PTSD and the 8 hypothesized associated features. The scale yields a dichotomous diagnosis of PTSD and also provides a continuous score regarding the frequency and severity of each symptom. The PTSD symptom clusters assessed include re-experiencing, hyperarousal, behavioral avoidance, and emotional numbing. This study used the FI/I2 scoring rule, in which a symptom is considered present if the frequency of the corresponding CAPS item is rated as 1 or higher and the intensity is rated as 2 or higher. The CAPS was scored based on the most traumatic event reported by the participant, and follow-up CAPS assessments focused on the same event. 

PTSD Checklist—Civilian Version (PCL-C) [24]: The PCL is a 17-item self-report questionnaire designed to assess PTSD symptomatology. Participants are presented with a list of PTSD symptoms and asked to indicate the extent to which they have been bothered by each during the past month using a 5-point Likert-type scale. Scores may range from 17 to 85. A number of studies have attempted to identify a PCL cutoff score with appropriate specificity and sensitivity to suggest further screening for PTSD [25]. For the purpose of the present study, a PCL-M cutoff of ≥50 was used to define “significant PTSD symptomatology”. The Civilian version was administered in order to assess for all sources of trauma. The PCL has demonstrated strong internal consistency and reliability over time and appears to be a valid measure of PTSD. In addition, this measure has demonstrated good sensitivity (0.82) and specificity (0.83).

Numerical Pain Rating Scale (NRS) [26]: The NRS is a psychometrically strong, 0 to 10 rating scale, with 0 indicating “no pain” and 10 indicating the “worst pain imaginable”. Participants were asked to “Rate your average pain over the past 2 weeks using a 0 to 10 scale (0 = no pain and 10 = worst pain imaginable)”. 

Beck Depression Inventory (BDI) [27]: The BDI is a 21-item self-report measure of depressive symptom severity that is used to assess the extent to which an individual currently exhibits or experiences each of the behaviors, thoughts, or affective features of depression. Using a 0 to 3 scale, for each item, respondents are asked to choose the statement in each group of statements that best describes the way that they have been feeling in the past week, including the present day. Scores range from 0 to 63 (0–10 = normal; 11–16 = borderline; 17–20 = mild; 21–30 = moderate; 31–40 = severe; 41–63 = extreme). The BDI yields a total score for depressive symptom severity, as well as 2 subscales including cognitive and somatic symptoms of depression. 

The West Haven Yale Multidimensional Pain Inventory (WHYMPI) [28]: The WHYMPI is a 52-item self-report questionnaire consisting of 12 subscales that have been demonstrated to be applicable across a variety of clinical pain conditions. The internal reliability coefficients of all WHYMPI scales range from 0.70 to 0.90; the test–retest reliabilities of these scales over a 2-week interval range from 0.62 to 0.91. Using a 0 to 7 scale, respondents indicate the extent to which a specific question or statement applies to them. Subscale values range from 0 to 6. For the purpose of the present evaluation, only the “Interference” subscale was examined. 

Pain Catastrophizing Scale (PCS) [29]: The PCS is a self-report measure that asks participants to reflect on past painful experiences and to indicate the degree to which they experience each of 13 thoughts or feelings when experiencing pain on a 5-point scale from 0 (not at all) to 4 (all the time). Validation studies have found the measure to be a reliable and valid measure of catastrophizing.

Coping Strategies Questionnaire—Revised (CSQ-R) [30,31]: The CSQ-R was used to evaluate the use of adaptive and maladaptive pain coping strategies. As noted previously, the revised questionnaire includes six subscales: distraction, catastrophizing, ignoring pain, distancing, coping self-statements, and praying. The subscales of the CSQ-R have been established through multiple studies, with the reliability of the individual subscales reported to be between 0.72 and 0.86 [32]. For the present study, the “catastrophizing” subscale (e.g., “I worry all the time about whether it will end”) was examined. 

Perceptions of Treatment Questionnaire (PTQ): The PTQ is a 17-item self-report questionnaire assessing treatment satisfaction and acceptability that has been used in previous psychological treatment development studies for chronic pain [15]. Respondents are asked to indicate the extent to which they agree with statements using a 0 to 8 scale (0 = not at all, 4 = somewhat, 8 = very much). Also included are 6 questions asking respondents to “Rate how helpful you found each of the 6 treatment sessions” using a 0 to 10 scale (0 = not helpful and 10 = extremely helpful).

#### 2.3.3. Procedures

Potential participants were screened by telephone to review the study eligibility criteria. Patients meeting the initial criteria were scheduled for an assessment by an evaluator. Following the provision of informed consent, participants were administered the SCID and CAPS first so as not to unnecessarily burden those who did not meet the study criteria. The evaluator was an advanced doctoral-level student in clinical psychology who was trained to administer all of the clinician-administered interviews. Evaluator training included the completion of training videos, consultation with other trained assessors, and participation in a CAPS supervision group. Participants exhibiting diagnostic eligibility then completed all self-report questionnaires. 

Prior to the first therapy session, the study therapists were provided with an overview by the evaluator of the most severe traumatic event reported by the participant and a description of their chronic pain problem(s) and other issues that could impact the therapy. The traumatic event described was the focus of the PTSD treatment. The PATRIOT Program was delivered by the PI and a second PhD-level therapist, both of whom had expertise in CBT for chronic pain and CPT for PTSD and were involved in developing the integrated pain and PTSD treatment. The therapists met weekly for supervision and to discuss treatment issues. Given the schedules of Veterans, it was often the case that therapy appointments were scheduled at different times during the week or after normal working hours. Several procedures were used during treatment to reduce potential attrition. First, the therapists were trained about the importance of building an early, strong alliance with the participants. Other strategies to reduce attrition included setting clear expectations for the participants by describing early (during the informed consent procedure) the demands that would be made of them to attend therapy sessions twice a week, to review the written materials (handouts) covered in treatment, to attend assessment sessions, and to provide follow-up assessment information. During treatment, the therapists provided consistent reinforcement (therapist praise) to the participants for regular attendance, homework completion, and progress. The therapists called participants who did not attend scheduled appointments to express concern and confirm future appointments. 

After treatment, the evaluator conducted a post-treatment assessment, which included the same self-report forms completed at baseline, as well as the PTQ. The post-treatment assessment was conducted within 1 to 2 weeks of the final therapy session. Participants were compensated USD 30 for completion of the pretreatment assessment and USD 40 for completion of the post-treatment assessment.

## 3. Results

### 3.1. Feasibility and Acceptability

All eight participants completed the full 3-week course of the PATRIOT Program. When homework assignments were not completed sufficiently, time was spent in the session completing and reviewing the assignment. When behavioral goals were not fully completed, time was spent processing the factors that interfered with goal completion and setting new goals. The participants seldom experienced problems attending treatment twice a week. In fact, the pace and sequence of the sessions seemed to create a level of therapeutic momentum noticed by both the patients and therapists. Moreover, the qualitative data obtained from the PTQ at post-treatment included statements such as “This has been great, you have given me some tools that I can really use”, “I’m doing things I haven’t done in a long time”, “I needed this”, and “It helped me to sort things out and manage my pain and PTSD”. Patients’ ratings of the helpfulness of the sessions were particularly high [(M = 7.2 on a scale of 0 (“not helpful”) to 10 “extremely helpful”)], with the highest ratings of perceived helpfulness reported for Session 1 (M = 8.2, SD = 2.4), Session 2 (M = 8.7, SD = 1.5), and Session 3 (M = 8.3, SD = 1.9).

### 3.2. Clinical Outcomes

Table 3 presents the individual and group-level clinical outcomes across participants. Despite the small sample size of this pilot evaluation, the Wilcoxon signed-rank tests identified significant improvements following the completion of the PATRIOT Program in terms of PTSD symptoms, average pain ratings, and pain catastrophizing (*p* < 0.05), and the improvements in depressive symptoms trended towards significance (*p* = 0.06). At the same time, across the brief 3-week study period, significant changes were not observed in pain-related interference, although the reductions were in a positive direction. The examination of the magnitudes of the change score effect sizes provided further positive initial support for the PATRIOT Program. Specifically, very large effects were found for PTSD symptoms (CAPS *r* = −0.74 PCL-C *r* = −0.71), average pain ratings (*r* = −0.71), and pain-related catastrophizing (PCS *r* = −0.84, CSQ *r* = −0.84); a large effect was found for depressive symptoms (*r* = −0.67); and a medium effect was found for pain interference (*r* = −0.42). Focusing specifically on the outcomes obtained based on the CAPS using the F1/I2 rule, the results indicated that participants 2, 7, and 8 no longer met the diagnostic criteria for PTSD at post-treatment. A closer examination of the symptom clusters that comprised the diagnosis of PTSD indicated that the only symptom cluster that was significantly different from pre- to post-treatment was re-experiencing (*p* = *0*.012), although the improvements in emotional numbing trended towards significance (*p* = 0.09) (see Table 4). The improvements in re-experiencing symptoms were very large in magnitude (*r* = −0.88), and the improvements in emotional numbing were large in magnitude (*r* = −0.59).

## 4. Discussion

Amidst the concerning backdrop of high rates of comorbid pain and PTSD among Veterans, the PATRIOT Program offers the first intensive treatment protocol for the integrated treatment of chronic pain and traumatic stress. Encouragingly, the results of this open pilot evaluation suggest that the intensive 3-week PATRIOT Program appears to be feasible to implement in the VA healthcare system and acceptable to patients. After accounting for two participants who failed to engage in treatment before even attending their first session, there were no treatment dropouts and all participants completed the post-treatment assessments. Several factors may have contributed to the low dropout rate. First, the manualized treatment was focused on delivering practical skills that the participants could use to manage their pain and PTSD. Second, the treatment sessions were scheduled flexibly to meet the needs of participants who had jobs, medical appointments, or other responsibilities. Third, the participants were reimbursed for their participation in the assessments. Importantly, however, the low dropout rate may have been the result of the novel intensive delivery format of the treatment. Seeing patients for therapy twice a week for 90 min allowed more time to quickly develop a therapeutic alliance with the patients and may have contributed to a greater sense of teamwork and commitment on the part of the therapist and patient. Participants’ ratings of treatment satisfaction were high. The first three sessions of treatment received the highest ratings of helpfulness by the participants. Session 3, entitled “Focused Cognitive Restructuring”, in which the content of the session was individually tailored to the needs of each participant, was rated the highest. The qualitative feedback obtained at post-treatment also suggested that the participants found the treatment helpful and that they learned skills to manage their pain and PTSD more effectively.

The participants in this open pilot evaluation of the PATRIOT Program exhibited significant reductions in PTSD, average pain, and pain catastrophizing and marginally significant reductions in depressive symptoms. In order to assess PTSD, this study included both the CAPS and the PCL, and, on both of these measures, the participants reported significantly reduced symptoms at post-treatment. On the CAPS, which is considered the gold-standard assessment for the diagnosis of PTSD, three participants no longer met the criteria for PTSD after just 3 weeks. On closer examination of the PTSD symptom clusters on the CAPS, the majority of the pre- to post-treatment change was concentrated in the re-experiencing symptom cluster. Re-experiencing symptoms of PTSD involves reliving a traumatic event in the form of intrusive thoughts, nightmares, dissociative flashbacks to elements of the original traumatic event (i.e., sights, sounds, or smells), and psychophysiological reactivity to cues of the traumatic event and preoccupation with that event. There is research suggesting that there may be a connection between the experience of chronic pain and re-experiencing symptoms of PTSD. A study by Powell et al. [33] found that, in a sample of Veterans with comorbid chronic pain and PTSD, the pain intensity was significantly associated with total PTSD score as measured by the CAPS; however, re-experiencing was the only PTSD symptom cluster that was significantly associated with pain. It was theorized that the experience of chronic pain, particularly when the pain was related to the traumatic event, could serve as a reminder of the event and increase the re-experiencing symptoms. Supporting this theory, a more recent study found that, in a sample of Veterans reporting trauma-related pain, the pain intensity was predictive of post-treatment re-experiencing symptoms [34]. 

One potential mechanism linking the experience of chronic pain and re-experiencing symptoms is catastrophizing. Research has shown that patients with chronic pain and high levels of PTSD symptomatology (i.e., scores greater than 50 on the PCL) score higher on measures of catastrophizing when compared to Veterans without significant PTSD symptomatology [9]. It is possible that the PATRIOT sessions that were focused on cognitive restructuring (Sessions 2 and 3) and nightmares (Session 4) helped the participants to challenge their catastrophic thinking and beliefs about controllability that were applicable to both the traumatic event and the experience of chronic pain. The fact that the participants reported significant reductions in the re-experiencing symptoms of PTSD and pain-related catastrophizing may suggest that their participation in the PATRIOT Program strengthened their adaptive coping and reduced their catastrophizing, which subsequently reduced their ratings on the measures of PTSD. 

Because of the brief nature of the treatment, the behavioral goals that were set in Session 1 were short-term goals, such as starting a walking program, going to the gym, or organizing a space in the home. The study results indicated that there were no significant changes from pre- to post-treatment in the extent to which pain was perceived as interfering in everyday activities. It is possible that the intensive nature of this treatment did not allow sufficient time for the goals to be fully accomplished or for increased activity related to weekly behavioral goals to be internalized as “less interference”. The participants were encouraged to continue to work on the behavioral goals after the therapy terminated. Perhaps continued follow-up and patient adherence to behavioral goal attainment would result in the participant’s perception of decreased pain interference. Future studies could include a stronger behavioral activation component that could be enhanced by the use of heart rate or activity monitoring devices. 

This study’s strengths include the development and use of a manualized treatment protocol for therapists and a patient workbook and the use of psychometrically sound assessment instruments specifically designed to assess pain and PTSD. There were several study limitations, including the lack of a treatment control condition and the small number of study participants in this preliminary pilot evaluation. An assessment limitation was the use of the NRS, with the instructions asking participants to rate their average pain over the past 2 weeks. Although the use of the NRS is recommended by the Initiative on Methods, Measurement, and Pain Assessment in Clinical Trials (IMMPACT) [35], the instructions generally ask the participant to rate their level of pain over the last 24 h or 1 week. Since the post-treatment assessment took place between 1 and 2 weeks after study completion, it is possible that, when completing the pain assessment, the participants were reflecting on their experience of pain that would include the final week of treatment. This study was conducted prior to the release of the DSM-5-based assessments; thus, both the CAPS and the SCID interviews were based on the DSM-IV. Another limitation was that the days between sessions were not evenly spaced. For example, for some participants, Session 1 started on a Thursday, whereas, for other participants, Session 1 started on a Monday. However, for all participants, we limited participation to 2 sessions per week. Future studies should include follow-up data collection. Controlled evaluations of the PATRIOT Program in adequately powered samples are now needed in order to formally examine the treatment effectiveness and generalizability.

Overall, the results of this study support the acceptability and feasibility of this novel intensive treatment approach in helping Veterans to learn skills to manage chronic pain and PTSD. With continued support, the PATRIOT Program could be implemented before pain or PTSD become chronic, could improve patient engagement, would be cost-effective, and could potentially reach a greater number of Veterans around the world. The PATRIOT Program could be considered a “first step” to engaging individuals in additional psychological treatment programs to help them to maintain the skills that they have learned or strengthen their skills to effectively cope with chronic pain and PTSD. Future controlled evaluations of the PATRIOT Program should seek to incorporate attention control conditions in order to account for the effects of therapists’ attention and treatment expectations. Moreover, although developed for a Veteran population, the skills taught in the PATRIOT Program are certainly applicable to all populations. Given the high rates of pain and PTSD among civilians, such as individuals who live in areas where there are higher rates of crime and violence or limited access to healthcare resources, evaluations of the PATRIOT Program could also examine patient outcomes among civilian populations with comorbid pain and PTSD.

## 5. Conclusions

The development and initial pilot evaluation of the PATRIOT Program is part of a growing body of work supporting the many advantages of brief, intensive treatment formats for difficult-to-reach and difficult-to-treat patient populations. The present findings provide promising support for the very positive role that such an innovative treatment format can offer Veterans suffering with both chronic pain and PTSD. 

## Figures and Tables

**Table 1 behavsci-14-01103-t001:** Pain and Trauma Intensive Outpatient Treatment (PATRIOT) Program.

Session	Title	Content
1	Making the Connection	Reviewing the connection between pain and PTSD, setting goals, and learning how to relax.
2	Cognitive Restructuring	Understanding the power of thoughts, learning to recognize thoughts that are not adaptive, and practicing how to challenge them.
3	Focused Cognitive Restructuring Anger Management Power/Control Trust/Safety	Restructuring that is tailored to the specific types of thoughts that are most problematic for a patient.
4	Sleep and Relaxation Training	Learning ways to improve sleep and address nightmares.
5	Activity Pacing and Pleasant Activities	Reintroducing pleasant activities into a person’s life and learning how not to overdo it on tasks.
6	Social Support and Integrating Skills into Everyday Life	Developing skills for creating a social support network and problem-solving how to deal with future pain and PTSD flare-ups.

**Table 2 behavsci-14-01103-t002:** Demographic characteristics of Veterans who participated in the open pilot evaluation of the PATRIOT Program (N = 8).

Participant	Age	Sex	Education (Years)	Race/Ethnicity	Military Duty	Relationship Status	Pain Site(s)	Pain Duration (Months)	Employment Status
1	68	M	15	Caucasian (non-Hispanic)	Non-combat	Divorced	Shoulder, back, knee	37	Retired
2	69	M	7	Caucasian (non-Hispanic)	Combat—Vietnam	Single	Hip, leg, back	360	Unemployed
3	63	M	12	Black (non-Hispanic)	Non-combat	Significant Other	Arm, shoulder, knee	50	Unemployed
4	33	M	12	Caucasian (non-Hispanic)	Combat support—OEF/OIF	Single	Hip, leg, knee	12	Unemployed
5	28	M	16	Caucasian (non-Hispanic)	Combat—OEF/OIF	Single	Shoulder, neck	60	Full-time
6	59	F	18	Native American	Non-combat	Divorced	Whole body	48	Full-time
7	60	M	16	Native American	Combat—Vietnam	Single	Foot, hip, leg, arm, shoulder, head, neck, back	49	Unemployed
8	49	M	14	Native American	Combat—Gulf War and OEF/OIF	Married	Shoulder, neck, back, knee, whole body	67	Part-time

**Table 3 behavsci-14-01103-t003:** Individual and group clinical outcomes, across time, among Veterans who participated in the open pilot evaluation of the PATRIOT Program (N = 8).

	CAPS		PCL-C		Average Pain Rating		WHYMPI Pain Interference		Pain Catastrophizing Questionnaire		CSQ-Catastrophizing		BDI	
Participant	Pre	Post	Pre	Post	Pre	Post	Pre	Post	Pre	Post	Pre	Post	Pre	Post
1	83	77	-	57	6	5	2.3	3.6	-	21	2.5	2.2	-	21
2	75	53	58	53	9	8	3.1	3.1	21	20	1.3	1.3	17	16
3	73	45	62	71	5	6	4.0	4.0	29	28	3.9	2.0	21	22
4	68	73	85	52	6	4	5.4	3.8	24	19	2.6	2.1	21	21
5	91	62	70	60	7	5	5.3	4.3	51	35	3.8	2.3	33	24
6	100	91	75	57	10	6	2.0	2.0	38	2	2.8	0.8	31	9
7	43	24	61	51	7	7	5.1	4.6	33	24	2.9	1.7	25	14
8	44	48	50	37	9	6	4.4	2.1	15	4	2.0	0.6	21	8
Mean	72.1	59.1	65.9	54.8	7.4	5.9	4.0	3.4	30.1	18.9	2.7	1.6	24.1	16.3
Median	74.0	57.5	62.0	55.0	7.0	6.0	4.2	3.7	29.0	20.5	2.7	1.8	21.0	18.5
SD	20.4	21.1	11.7	10.3	1.77	1.25	1.4	1.0	12.0	12.1	0.9	0.7	5.9	6.3
Wilcoxon Signed-Rank Test	*Z* = −2.1, *p* = 0.036		*Z* = −2.0, *p* = 0.03		*Z* = −2.0, *p* = 0.041		*Z* = −1.2, *p* = 0.25		*Z* = −2.4, *p* = 0.018		*Z* = −2.4, *p =* 0.017		*Z* = −1.9, *p* = 0.06	

**Table 4 behavsci-14-01103-t004:** Individual and group clinical outcomes on the clinician-administered assessment of PTSD (CAPS).

	Total Score		Re-Experiencing		Hyperarousal		Behavioral Avoidance		Emotional Numbing	
Participant	Pre	Post	Pre	Post	Pre	Post	Pre	Post	Pre	Post
1	83	77	33	27	18	16	16	12	16	22
2	75	53	30	19	22	21	5	11	18	2
3	73	45	16	11	20	17	10	4	27	13
4	68	73	21	17	23	26	13	13	11	17
5	91	62	19	17	31	21	12	9	29	15
6	100	91	32	31	29	19	13	12	26	29
7	43	24	14	7	17	14	4	2	8	1
8	44	48	13	10	17	28	4	10	10	0
Mean	72.1	59.1	22.3	17.3	22.1	20.3	9.6	9.1	18.1	12.4
Median	74.0	57.5	20.0	17.0	21.0	20.0	11.0	10.5	17.0	14.0
SD	20.4	21.1	8.2	8.3	5.3	4.8	4.7	4.0	8.3	10.6
Wilcoxon Signed-Rank Test	*Z* = −2.1, *p* = 0.036		*Z* = −2.5, *p* = 0.012		*Z* = −0.85, *p* = 0.40		*Z* = 0.34, *p* = 0.73		*Z* = −1.68, *p* = 0.09	

## Data Availability

The data generated and/or analyzed in the present study are available from the corresponding author on reasonable request.

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
