# Peer review of "Intensive Treatment of Chronic Pain and PTSD: The PATRIOT Program"

_behavsci, 2024, doi:10.3390/bs14111103_

Round 1
Reviewer 1 Report
Comments and Suggestions for Authors
The original research titled "Intensive Treatment of Chronic Pain and PTSD: The PATRIOT Program" methodically addresses an important area, offering a proof of concept for this intensive treatment model.
• Line 37: Expand "OEF/OIF" when first mentioned, as readers outside the US may not be familiar with these terms.
• Line 294: For clarity, provide the number of male and female participants in whole numbers, rather than stating “87.5% of the sample.” Additionally, correct the sentence by removing the unnecessary "and" in “age of participants was and 53.6.”
• Line 422: The scales used to measure outcomes appear to be based on ordinal values rather than continuous data. Therefore, I have concerns about the use of paired t-tests in this analysis for the following reasons:
1. Suitability of t-tests: Using t-tests (including paired t-tests) for ordinal data is methodologically questionable, despite being a common practice.
2. Normality Assumption: Paired t-tests assume a normal distribution, which is a potentially flawed assumption given the small sample size in this study. Non-parametric alternatives like the Mann-Whitney U test or Wilcoxon Signed-Rank test would have been more appropriate. Nonetheless, these options still pose the issue of applying tests designed for continuous data to ordinal outcomes.
3. Recommendation: A chi-square test might have been a more suitable choice. If the authors decide to retain the paired t-test approach, they should provide a clear justification.
Until these methodological issues are clarified, it is challenging to interpret the study's results. However, the conclusions appear reasonable based on the presented data.
Author Response
Reviewer 1 Comments
Comment 1: Expand "OEF/OIF" when first mentioned, as readers outside the US may not be familiar with these terms.
Response: We have made that change in the manuscript.
Comment 2: For clarity, provide the number of male and female participants in whole numbers, rather than stating “87.5% of the sample.” Additionally, correct the sentence by removing the unnecessary "and" in “age of participants was and 53.6.”
Response: We have removed the unnecessary “and” from the sentence. In addition, we have modified the sentence to improve readability “Participants consisted of 7 males and one female; the average age of participants was 53.6 (SD=18.3).
Comment 3: Suitability of t-tests: Using t-tests (including paired t-tests) for ordinal data is methodologically questionable, despite being a common practice. Normality Assumption: Paired t-tests assume a normal distribution, which is a potentially flawed assumption given the small sample size in this study. Non-parametric alternatives like the Mann-Whitney U test or Wilcoxon Signed-Rank test would have been more appropriate. Nonetheless, these options still pose the issue of applying tests designed for continuous data to ordinal outcomes. Recommendation: A chi-square test might have been a more suitable choice. If the authors decide to retain the paired t-test approach, they should provide a clear justification.
Response: In the revised report, we now evaluate the significance of pre- to post-treatment changes with Wilcoxon signed-rank tests, as we agree these are more appropriate and compelling in light of the nature of the data and the small sample size. (note: chi square tests would not have been appropriate, as such tests are geared toward categorical data, and require an assumption of independence of observations—whereas these are within-subjects, paired and repeated data). In addition to replacing all t-tests with Wilcoxon signed-rank tests, we also now report effect sizes as r’s to correspond with Wilcoxon signed-rank tests (d’s are not appropriate in the context of non-parametric analyses). We have also added data on medians in addition to means for all variables, as these values are used in the Wilcoxon signed rank test. This new analytic approach did not alter the interpretation of findings, with the exception that improvements in depression symptoms now trend toward significance (p=.06) whereas before these improvements reached the significance threshold (previous p=.048). In addition, improvements in emotional numbing now trend toward significance. Language throughout the manuscript has been adjusted to appropriately characterize these results.
Reviewer 2 Report
Comments and Suggestions for Authors
I think this is a well written paper worthy of publication. It is original in that it seeks to combine Pain and PTSD treatment in one brief program. It does advance the field.
The only issue I had was why use DSM-IV criteria not -V. This was raised in the discussion. I think the explanation in line 520 could be made clearer and reasons better explained. Otherwise I have no issue with this paper being published as it has been written.
Author Response
Reviewer 2 Comments
Comment 1: The only issue I had was why use DSM-IV criteria not -V. This was raised in the discussion. I think the explanation in line 520 could be made clearer and reasons better explained.
Response. We agree that we should have been clearer. The explanation for why we used measures based on DSM-IV and not DSM5 has been modified to reflect that the study was conducted before the release of the DSM5-based assessments.
Round 2
Reviewer 1 Report
Comments and Suggestions for Authors
Thank you for making the changes. your comment about the chi-square test is well made.